# Expanding Indications in Transplant Oncology

**DOI:** 10.3390/cancers17050773

**Published:** 2025-02-25

**Authors:** Erlind Allkushi, Chase J. Wehrle, JaeKeun Kim, Mazhar Khalil, David C. H. Kwon, Masato Fujiki, Antonio D. Pinna, Charles Miller, Andrea Schlegel, Federico Aucejo, Koji Hashimoto, Alejandro Pita

**Affiliations:** Transplantation Center, Department of Liver Transplantation, Cleveland Clinic, Cleveland, OH 44195, USAwehrlec@ccf.org (C.J.W.); kimj30@ccf.org (J.K.); khalilm5@ccf.org (M.K.); kwonc2@ccf.org (D.C.H.K.); fujikim@ccf.org (M.F.); pinnaa@ccf.org (A.D.P.); schlega4@ccf.org (A.S.); hashimk@ccf.org (K.H.)

**Keywords:** liver transplantation, transplant oncology, HCC, CRLM, CCA

## Abstract

Transplant oncology, or liver transplant as a curative-treatment for liver cancer, is increasingly utilized and can provide excellent outcomes. This has been done for over three decades in the treatment of hepato-cellular carcinoma (HCC), though evidence continues to mount in support of expanding the pool of candidates with HCC. Non-HCC liver tumors, however, are increasingly treated with transplantation, which represents a novel approach to provide curative-intent therapy to more and more patients. This article reviews indications for this approach and available data about outcomes of liver transplant as an oncologic treatment for different cancer types.

## 1. Introduction

Liver transplantation (LT) has become increasingly accepted as a curative intent treatment option for unresectable hepatic malignancies. As recently as 2021, hepatocellular carcinoma (HCC) and perihilar cholangiocarcinoma (pCCA) were the only hepatic malignancies widely treated with liver transplantation [1]. A comprehensive review of the literature in 2023 revealed the increased utility of liver transplantation with curative intent for colorectal cancer liver metastasis (CRLM), primarily in patients with low a Oslo and Fong Clinical Risk Score (FCRS) [2]. Further, interest has increased in the treatment of intrahepatic cholangiocarcinoma (iCCA) with LT due to the overall poor prognosis of this disease. Recent publications point to the importance of tumor biology, responses to therapy, and long-term disease stability in cases with favorable outcomes [3]. With the wider application of locoregional therapies (LRT) and downstaging treatments, as well as the increasing percentage of patients with unresectable hepatic malignancies, the discussion of expanding liver transplant oncology indications is pertinent. This article aims to provide a review of currently accepted indications for liver transplantation, as well as a summary of recent publications on expanded indications with positive outcomes. Our hope is to foster discussion on new oncological treatment protocols that may increase access to transplants with potentially curative intent.

## 2. Hepatocellular Carcinoma

Liver transplantation has been shown to be the best treatment option for early-stage hepatocellular carcinoma (HCC). In 2023, HCC was the primary diagnosis for 10.3% of liver transplant waitlist candidates, and 10.4% of transplant recipients in the U.S. alone [4]. The most common causes of liver cirrhosis are metabolic dysfunction-associated liver disease (MASLD), formerly known as non-alcoholic steatohepatitis (NASH), and alcoholic (ETOH) cirrhosis [5,6]. Unsurprisingly, these are also the two leading etiologies of liver disease that precede HCC [5]. There are treatments for early-stage MASH that can mitigate the risk of progression, but advanced MASH is increasing in both prevalence and severity and presenting at earlier stages than ever before [7,8]. Even more critically, the projected incidence of MASH in the next 30 years is >40%, which would in turn lead to a projected doubling of the incidence of liver cancer and a quadrupling of the overall need for liver transplantation [9]. ETOH consumption is clearly linked in a dose-dependent fashion to liver damage, cirrhosis, and, in turn, the development of HCC [5]. While heavy consumption is most commonly associated with cirrhosis, any degree of ETOH usage may predispose patients to liver cancer. Interestingly, the degree of alcohol intake is interdependently linked with metabolic risks such as diabetes, age, obesity, and dysbiosis, highlighting how ongoing and dys-synergistic liver damage predispose patients to oncologic transformation [5,10]. The rising burden of MASH and the stable burden of ETOH cirrhosis make the treatment of HCC increasingly relevant. These etiologies further highlight the need for transplantation, as these conditions are themselves best treated curatively by transplantation regardless of the presence of HCC, thus allowing one treatment to address both issues.

Historically, liver transplantation in the setting of HCC has been guided by the Milan criteria [11]. However, later publications have pointed to the potential limitations of these criteria due to tumor size restrictions. Data from UCSF showed 1- and 5-year survivals of 90% and 75.2%, respectively, in a cohort of 70 patients who received transplants with lesions. Analysis employed expanded parameters beyond the Milan criteria (tumors ≤ 6.5 cm or ≤3 nodules with the largest lesion ≤ 4.5 cm and total tumor diameter ≤ 8 cm) [12]. Additionally, Kyoto criteria incorporated tumor markers for patients with up to 10 tumors (each no more than 5 cm in diameter) at the time of transplantation, with a 5-year OS of 87% included in these criteria [13]. Furthermore, poor tumor differentiation and microvascular invasion are emphasized by both the Kyoto and Milan groups as contraindications for transplant due to the high risk of recurrence. The application of LRT protocols prior to LT led to the development of expanded downstaging USCF criteria, in addition to the Milan criteria. Locoregional therapy has been shown to improve outcomes in HCC patients within Milan and UCSF criteria when tumors demonstrated a complete pathologic response (cPR) pretransplant [14,15].

In 2018, the Milan group introduced the Metroticket 2.0 Model, which emphasized the predictive significance of tumor size, tumor number, and AFP levels regarding survival and recurrence risk in HCC patients [16]. Further validation reported 3- and 5-year OS values of 88.6% and 79.1%, respectively. Of note, HCC recurrence increased in patients with higher AFP values, as well as in those patients not treated under Milan criteria who underwent chemoembolization [17,18,19].

In 2024, comprehensive analysis, comparing multiple selection criteria for LT in HCC treatment, was performed (SRTR national database n = 26,409; Milan & UCSF n = 547) [17]. Transplant candidacy and outcomes (3-year OS) were compared among Milan, UCSF, 5-5-500, U7, Metroticket 2.0, and HALT-HCC criteria. Results showed that Metroticket 2.0, UCSF, and U7 criteria could increase transplant utilization for patients with HCC while maintaining positive outcomes. It was also noted in this analysis that HALT-HCC and Metroticket 2.0 were the best criteria for predicting recurrence. For this reason, the expansion of criteria beyond the Milan variant is gaining traction, with Metroticket 2.0 shown in this study to be both the most discriminative and least restrictive criteria, yet also able to maintain post-LT outcomes [17].

Downstaging has led to comparable survival results for HCC patients treated within and outside of Milan and UCSF criteria. Some centers base patient eligibility on disease stability over 6–9 months, as well as on responses to LRT. Washington University reported successful downstaging for 63/210 patients with HCC who made it to transplantation [20]. Recurrence-free survival (RFS) for downstaged patients was shown to be similar to that of those treated within Milan and UCSF criteria. HCC recurrence was also reported to be similar between groups, with 8.9% for the downstaged group, 5.6% for the UCSF group, and 9.2% for the group initially within Milan. Expanded indications may support patients with favorable tumor biology, disease stability over time, and a complete pathologic response (cPR) to LRT pretransplant [15]. Despite concerns of risk of recurrence, aggressive posttransplant LRT and surgical treatments appear to provide some benefit in terms of long-term survival.

Combination treatments with LRT and immunotherapy show promising outcomes [21]. Ablation techniques such as radiofrequency ablation (RFA) and microwave ablation (MWA) show comparable outcomes to resection for well-defined and low-grade HCC tumors, though both present some limitations. Histotripsy, a non-invasive, non-thermal, and non-ionizing ablation technique, has recently emerged as a potential downstaging treatment [22]. Though data are limited, early-stage reports show promising abscopal effects using this technique. Yttrium-90 (Y-90) trans-arterial radioembolization (TARE) has also emerged as a very useful downstaging and bridging modality for all stages of HCC, demonstrating an excellent safety profile and having the additional benefit of inducing future liver remnant growth when resection is being considered [23]. Radiotherapy can be combined with novel immunotherapies such as atezolizumab/bevacizumab or ipililumab/nivolumab in possibly synergistic ways, though this is as much speculative as fully elucidated [21]. With regard to combination treatments, a review by Kumar et al. found that a combination of LRT with sorafenib, atezolizumab–bevacizumab, or Lenvatinib has shown potential OS benefits, but these techniques require further refinement due to frequent adverse events [21].

The post-liver transplantation management of HCC patients is exceptionally challenging. Though immunotherapies are primary treatments for HCC recurrence after resection, these have been historically contraindicated posttransplantation due to the immunologic inducement of rejection. The greatest risk seems to be secondary to PD-1/PD-L1 inhibitors, with a much lower rate described with CTLA-4 inhibitors [24]. However, there is evidence that use of IO post-solid organ transplants may be both possible and effective [25]. This is one area where further trials may have a critical impact on future management, just as non-transplant-related studies and trials on immunotherapy have led to extensive improvements in outcomes [26,27,28,29]. Finally, multicenter evidence does support direct liver resection after liver transplant as having an improvement on OS, meaning resection should be pursued in cases of resectable disease [30].

In summary, the management of HCC remains difficult due to nonresponse to cytotoxic chemotherapy, resistance to systemic immunotherapy, as well as the variability of disease burden among patients. The Milan selection criteria have historically proven successful in terms of outcomes for LT in HCC treatment. However, new transplant selection criteria for this cohort are emerging that could increase access without compromising outcomes. It is of the utmost importance that these criteria are validated to increase access to this lifesaving treatment. Emerging data on new LRT modalities such as histotripsy and combination treatments with LRT and immunotherapies have shown survival benefits, showing promise for downstaging HCC patients within acceptable selection criteria. In light of these recent developments, new guidelines ought to be established soon to apply pretransplant LRT or combination therapies, as well as to expand selection criteria beyond the currently used, albeit outdated, Milan criteria.

## 3. Colorectal Liver Metastasis

Liver metastasis manifests in at least 25% of patients with metastatic colorectal cancer during the course of their illness [31]. Currently, liver resection remains the gold standard for resectable CRLM, with 5- and 10-year OS values of 44–50% and 24–33%, respectively [32]. The introduction of a hepatic artery infusion pump (HAIP) after liver resection (LR) has also improved the 10-year OS rate to 38% [32,33]. However, 40–50% of CRLM cases are unresectable upon diagnosis, and management with chemotherapy has led to poor outcomes (5-year OS < 10%) [34].

Liver transplantation for unresectable CRLM was most notably supported by the SECA I trial out of Oslo University, which reported 1- and 5-year OS values of 95% and 60%, respectively [35]. These findings were validated by multiple centers showing OS values between 89 and 100% 1-year post-LT, though RFS was highly variable across each center (25–60%) [36]. This pilot study led to the development of the Oslo score, which is now a commonly used predictor of recurrence (Table 1) [2]. The Oslo score is based upon four criteria, with the manifestation of each criterion posing increased risk of recurrence for patients with CRLM. The SECA I trial reported significantly improved OS values in patients with Oslo ≤ 1 (10-year OS 88.9%, median OS 92.0 months) compared to patients with Oslo ≥ 3 (10-year OS 0%, median OS = 24.8 months) in the setting of unresectable CRLM. Overall survival (OS) in the setting of unresectable CRLM significantly improved with liver transplantation as opposed to resection for patients with an Oslo score ≤ 2, both in the short term and during long-term follow-up (69.1% vs. 14.6%, respectively). This finding is well complemented by data from Dueland et al., who reported poor long-term outcomes in patients with Oslo scores of 3–4 [35,36]. However, the SECA-I trial had limitations, most notably the lack of a control group. Even with the more stringent selection criteria of the SECA-II trial showing improved OS scores in 1, 3, and 5-year follow-ups (100%, 83%, 83%, respectively), this limitation remained [37].

In 2024, Adam et al. published the results of TRANSMET, a multicenter, prospective randomized trial comparing outcomes of patients with unresectable CRLM undergoing chemotherapy (Group C) versus chemotherapy plus LT (Group LT + C) (n = 94, 1:1 randomization) [38]. In intent-to-treat analysis, the 5-year OS for patients undergoing chemotherapy plus LT was 57%, compared to 13% in the chemotherapy-alone group. In per-protocol analysis, the 5-year OS was 73% versus 9%, respectively. This trial also showed an improved progression-free survival (PFS) in patients receiving chemotherapy plus LT versus chemotherapy alone (17.4 months vs. 6.4 months). The selection criteria for this trial are listed in Table 2 (CT.gov NCT 02597348). These results reflect once again the strong potential of LT in treating unresectable CRLM. However, further trials are warranted to determine the proper pre- and posttransplant therapies to improve poor DFS posttransplant [35].

Another critical discussion is the management and selection of potential LT candidates. In addition to the Oslo score validation, Dueland et al. also demonstrated improved OS in patients with pretransplant PET-MTV < 70 cm^3^ [39]. The combined groups from Cleveland Clinic and University of Rochester recently validated MTV as a selection criterion, providing further confirmation that lower MTV values can maximize outcomes [40]. Most notably, 1- and 2- year OS values were 100% and 85%, respectively, in the low-PET-MTV groups.

The Cleveland Protocol of 2024 established that pre-liver transplant (pre-LT) detectable disease serves as a predictor of recurrence in patients with CRLM [41]. The protocol described a novel pre-liver transplant plan of care focused on locoregional therapy and systemic chemotherapy to reduce disease burden prior to transplantation. Locoregional therapy included Y90, ablation, resection, and HAIP, while systemic therapy included chemotherapy +/− bevacizumab and/or anti-EGFR. Data on eligible candidates (n = 16) showed that long-term systemic therapy (minimum 6 months) and locoregional therapy with repeat evaluation every 3–6 months led to improved survival when compared to patients who were not candidates (n = 11).

Finally, selection criteria may continue to expand the incorporation of personalized medicine approaches, such as tissue-based methods or the use of circulating tumor DNA (ctDNA) [42,43]. These approaches are new but demonstrate potential in both surveillance and selection settings.

## 4. Cholangiocarcinoma

Cholangiocarcinoma (CCA) originates from cholangiocytes of the bile duct and carries a high risk of recurrence. It is the second leading cause of primary liver malignancy, and a common cause of death in patients with primary sclerosing cholangitis [44]. CCA currently manifests as intrahepatic, hilar, and distal. Unresectable perihilar CCA (pCCA) has become an accepted indication for liver transplant [1,44]. Similar to HCC, the other common primary liver tumors, MASH and ETOH, are the most commonly associated etiologies that precede CCA [45,46,47]. As such, it would be optimal to manage CCA similarly using liver transplantation; however, it has proven to be oncologically riskier than its more common counterpart.

Due to its infiltrative nature, the diagnosis of pCCA remains a challenge in most cases. The ILTS Transplant Oncology consensus guidelines recommended diagnosis using the following criterion—the presence of a dominant perihilar bile duct stricture—and one or more of the following criteria: positive cytology by endoscopic brushing or biopsy, fluorescence in situ hybridization (FISH) polysomy, or elevated CA 19.9 > 100 U/mL in the absence of cholangitis [48]. Since protocols were introduced, including neoadjuvant therapies (including different modalities of this, such as chemotherapy, external beam radiation, and/or brachytherapy), liver transplantation has resulted in significantly improved outcomes, with data showing overall survival rates at 5 years post-LT of 51–68%, and up to a 65% 5-year RFS [49,50,51]. As a result, consensus guidelines only recommend LT in patients with unresectable disease after neoadjuvant chemoradiation in centers with specific protocols [48].

The treatment of intrahepatic CCA (iCCA) remains controversial with regard to liver transplants, although recent studies are increasingly shedding light on appropriate patient selection and options for pretransplant treatments. iCCA represents 10–20% of all CCA tumors, with poor OS values of 10–40%. Small-size tumors, well-differentiated tumors, and tumors without lympho-vascular invasion are shown to have best outcomes [3]. While surgical options are limited due to the aggressive nature of the disease, medical therapy and LRT have shown benefit in RFS. While resection is the first-line treatment, unresectable tumors, such as those arising in the presence of cirrhosis, portal hypertension, or other forms of chronic liver disease, may benefit from LT. The treatment of iCCA with LT showed poor survival and high recurrence rates of >50% during initial studies. A retrospective analysis from Sapisochin et al. indicated that patients with solitary nodules ≤ 2 cm in diameter have comparable 5-year OS values to HCC patients after transplantation (62% vs. 80%, respectively) [52]. These results were supported by the Mayo Clinic, specifically for early-stage iCCA without vascular invasion, with reported a 5-year OS value of 63.6% compared to the 70.3% value seen for patients with HCC treated under the Milan criteria [53]. These findings were further validated in multiple studies, including a meta-analysis of eighteen studies, including 355 cases, which showed that in patients with very early (≤2 cm) iCCA, the pooled 5-year RFS was 67% [54]. An international study comparing liver transplantation outcomes in very early (≤2 cm) vs. more advanced disease (>2 cm) showed 1-, 3-, and 5-year actuarial survival rates of 91%, 84%, and 65% versus 79%, 50%, and 45% [55], suggesting that while advanced disease was associated with inferior outcomes, further studies are needed to elucidate if the 2 cm diameter cut-off may be too conservative.

The aforementioned studies did not explore the effects of neoadjuvant therapies pretransplantation, but UCLA data show improved survival with neoadjuvant therapy pretransplant for patients with CCA [56]. The Houston Methodist and MD Anderson published data on a cohort of 18 LT recipients who were required to demonstrate 6-month disease stability under neoadjuvant chemotherapy, showing post-LT OS values of 100%, 71%, and 57% at 1, 3, and 5 years, with a median number of 2 iCCA tumors and a median cumulative tumor diameter of 10.4 cm [57]. This study therefore emphasized that the tumor response to pretransplant treatment may be a more important criterion for survival than tumor size in the case of iCCA. In the expert consensus from the ILTS Transplant Oncology working group, recommendations were summarized as follows: liver resection remains the treatment of choice for resectable iCCA, while LT is reserved for unresectable cases and performed under strict protocols; patients with very early iCCA (single tumor ≤ 2 cm in diameter) in a cirrhotic liver would benefit from upfront LT, while patients with more advanced unresectable iCCA in a noncirrhotic liver may benefit from neoadjuvant therapy, reserving LT only for patients demonstrating stable disease [48]. Protocols continue to evolve as more promise is shown with the implementation of treatment modalities prior to liver transplantation.

## 5. Neuroendocrine Tumors

Neuroendocrine tumors (NETs) are rare, dormant neoplasms that primarily metastasize to the liver through the portal venous system. NET liver metastasis occurs at a rate of 40–93%, with a lower frequency in bones and lungs [58,59]. The OS at 5 years for metastatic liver NETs is 20–40%. Given the variability of presentation and symptoms, patients can often be unaware of the presence of NETs until metastatic disease reaches advanced stages. Metastatic liver NETs can be treated with a combination of RFA, TACE, chemotherapy, and resection.

However, for unresectable metastatic liver NETs, orthotopic liver transplants remain the best plan of treatment. After LT, the 5-year OS and DFS were shown to be 52% and 30%, respectively, for well-differentiated metastatic liver NETs (histologic grade 1–2), compared to 27% for poorly differentiated tumors (histologic grade 3) (ELTR, n-213) [60]. The European study also identified common predictors of poor outcomes to be hepatomegaly, age ≥ 45 years, and any amount of resection concurrent with LT. Other criteria have also been shown to affect the prognostic outcomes of OLT with regard to NETs, including the rate of tumor invasion, a Ki-67 index ≥ 5%, the number of tumors, and the presence of portal venous drainage [58,61]. The Milan–NET criteria demonstrated significant outcomes when used to determine LT recipients in this cohort [59]. Mazzaferro et al. reported in 2016 that metastatic liver NET patients within the Milan–NET criteria who were transplanted (n = 42) had 5- and 10-year survival rates of 97.2% and 88.8%, respectively, compared to values of 50.9 and 22.4% in non-transplanted patients (n = 46) [62]. The Milan–NET criteria also showed strong advantages with regard to time to progression, with values of 13.1% vs. 83.5% in the 5-year follow-up and 13.1% vs. 89% in the 10-year follow-up. A summary of Milan–NET criteria is included in Table 3.

Multiple studies have observed that LT can be also beneficial for patients with metastatic liver NETs when selection criteria are not highly restrictive [60,61]. A retrospective review by Nguyen et al. noted that since the introduction of the MELD score, OS values for metastatic liver NET patients improved significantly in the 1-, 3-, and 5-year follow-up compared to the pre-MELD era [63]. As previously noted, MELD exception criteria play a significant role in achieving higher access to patients where LT could improve OS and RFS. As a result, UNOS developed non-standard MELD exception guidance for neuroendocrine liver metastases patients who would benefit from LT, including bilobar disease confined to the liver and not amenable to resection; tumors with portal drainage system; tumor metastatic replacement not exceeding 50% of the total liver volume; an intermediate grade by WHO classification (well-differentiated/G1 and moderately differentiated/G2 tumors); and a mitotic rate < 20 per 10 HPF, with less than 20% Ki67 marker positivity [64]. Another potential predictor of survival is the wait time prior to transplant. Gedaly et al. reported improved patient survival when allowing for disease stability prior to LT for longer than 2 months [65]. A systematic review of the literature also supports the survival benefits of LT in patients with metastatic liver NETs, advocating for stricter selection criteria [66].

Further investigations are needed to support this treatment option, with better-refined selection criteria, such as downstaging and implementation of a pretransplant monitoring period.

## 6. Other Malignancies

The role of liver transplantation has also been explored in rare tumors of the liver, including hepatic epithelioid hemangioendothelioma (HEHE), hepatoid adenocarcinoma (HAC), and fibrolamellar carcinoma (FLC). HEHE affects approximately one in one million people worldwide [67]. Tumor involvement is present in the liver in 30% of cases. Other sites of infiltration include the lungs, peritoneum, abdominal lymph nodes, bone, and spleen, albeit at a lower rate [67,68]. Liver resection has led to the best outcomes for resectable HEHE cases, specifically 100% 1-year survival and 75% 5-year survival [67]. However, over 80% of patients present with bilobar involvement, excluding surgical resection as a treatment option [69]. Liver transplantation has the best outcomes and is the supported treatment modality for unresectable HEHE patients, while data remain scarce on the effectiveness of other therapies. One review of 434 primary HEHE cases reported that 44.8% of patients were treated with transplantation, with 1- and 5-year survival rates 96% and 55.4%, respectively [67]. The same study reported 1- and 5-year survivals of 73.3% and 30%, respectively, after treatment with chemotherapy or radiotherapy. UNOS data from 2002 to 2018 identified 131 adults listed for LT with an indication of HEHE [68]. All told, 88 of 131 patients underwent transplantation, with the 1-, 3-, and 5-year OS values reported as 88.6%, 78.9%, and 77.2%, respectively. Though we recognize the rarity of HEHE and subsequently the scarcity of the data available, current reports point to transplantation as an effective treatment plan [70,71]. Further studies are required to determine the effectiveness of other therapies and the best oncological treatment plans.

Fibrolamellar carcinoma (FLC) is another rare primary liver cancer that predominantly affects adolescents and young adults without underlying liver conditions [72,73]. The disease was thought to be a subtype of HCC, accounting for roughly 1% of all cases. However, recent studies point to clear distinctions between FLC and HCC tumors, suggesting that they should be treated as separate entities [74]. Such distinctions, including genomic and histologic alterations of FLC tumors, raise questions regarding the valid number of historic FLC cases and, subsequently, whether the proper prioritization and treatment plans are being implemented. Currently, surgical resection is the primary treatment modality for resectable FLC (5-year OS 76%), while the outcomes of unresectable FLC cases remain poor (5-year OS 10–20%) [75]. The predominant surgical resection treatment is major hepatectomy (70% of cases), with data pointing to significantly improved outcomes when complete resection (R0) is performed [76]. However, disease recurrence remains high following resection, with one study reporting a recurrence rate as high as 86% [77]. Some studies have reported improved OS when resection is followed by chemotherapy (median OS 23.1 months); however, data remain limited on the best systemic treatment modality [78]. Liver transplantation has shown itself to be a viable treatment option for unresectable FLC. Multiple large-scale retrospective studies have reported 5-year OS values of 48–55%, with a mean OS of 47.5 months [79,80,81]. However, OS is dependent on multiple factors such as tumor size, number, lymph node involvement, vascular invasion, and the utilization of systemic therapy. It is important to further explore liver transplantation for unresectable FLC in the prospective setting to establish best practices for treatment.

## 7. Conclusions

Hepatocellular carcinoma and hilar cholangiocarcinoma have become accepted indications for liver transplantation with curative intent. As more studies are emerging on the expansion of oncologic indications for liver transplantation, it is becoming increasingly clear that tumor biology and responses to pretransplant therapy are key factors for optimal oncologic outcomes. In addition, disease stability over time portends better outcomes post-operatively. More studies continue to support downstaging via locoregional therapy to bring patients’ treatment to within acceptable criteria for hepatic malignancies. Novel LRT treatments such as histotripsy are emerging; however, data remain limited on these approaches. It is worth exploring the importance of adjuvant and neoadjuvant therapies during pretransplant treatment, especially in cases of iCCA and HCC-CCA, as data are limited on overall survival when these therapies are applied in combination with liver transplants.

## Figures and Tables

**Table 1 cancers-17-00773-t001:** Oslo score criteria.

Oslo Score
Largest tumor size > 5.5 cm
Progressive disease at time of LT
Pre-operative CEA > 80 ug/L
Less than 2 years from primary tumor resection and liver transplant

**Table 2 cancers-17-00773-t002:** TRANSMET selection criteria.

Inclusion Criteria	Exclusion Criteria
≥18 and ≤65 years	Participation refusal
Good performance status, ECOG 0 or 1 (39).	No health insurance facilities
Histologically proved adenocarcinoma in colon or rectum	General contraindication to LT (severe cardiopulmonary disease or other life-limiting coexisting medical conditions, extrahepatic malignancy, active alcohol or substance abuse, active infection or uncontrolled sepsis, lack of psychosocial support, or inability to comply with medical treatment)
BRAF wild-type CRC on primary tumor or liver metastases	Other malignancies either concomitant or within 5 years before liver transplantation
High-standard oncological surgical resection of the primary defined as follows: ○ Safe margin of resection; ○ Curative resection of primary tumors according to oncological principles; ○ TNM adequate staging.	Patients not having received standard treatment for the primary CRC according to recommended guidelines
Absence of local recurrence on colonoscopy performed in the 12 months prior to inclusion (except in case of primary tumor resection < 12 months)	Prior extrahepatic metastatic disease or local relapse
Confirmed non-resectable colorectal liver metastases by the validation committee	Pregnancy at the time of inclusion
≥3 months of tumor control during the last chemotherapy line: stable or partial response based on RECIST criteria (40)	
≤3 lines of chemotherapy for metastatic disease	
CEA < 80 microg/L or a decrease ≥ 50% of the highest serum CEA levels observed during the disease	
Absence of extrahepatic tumor localization according to CT scan and PET-CT	
Renal function should be within the normal limits	
No need for extra-renal purification procedure, hemodialysis, or kidney transplantation-associated treatment (nephrologist assessment)	
A platelet count > 80,000/mm3	
White blood cell count > 2500/mm3	
Eligible for both treatments groups	
Signed informed consent and expected cooperation of the patient for the treatment and follow-up	

https://www.clinicaltrials.gov/study/NCT02597348 (accessed on 10 January 2025).

**Table 3 cancers-17-00773-t003:** Milan–NET inclusion criteria.

Confirmed Histology of Low-Grade (G1/G2 Grading According to the World Health Organization Classification)
Neuroendocrine tumor
Primary tumor removed with curative resection
Metastatic diffusion to liver parenchyma ≤ 50%
Stable disease for at least 6 months before liver transplant
Age ≤ 55 years

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
