# Peer review of "Expanding Indications in Transplant Oncology"

_cancers, 2025, doi:10.3390/cancers17050773_

Round 1
Reviewer 1 Report
Comments and Suggestions for Authors
This literature reviw provides a summary of the literature on treatment protocolsthat may treatment of for HCC with liver trans plantation that is timely and well done. However there is now evidence that non-alcoholic steatohepatitis , which is a misnomer, is the most common cause of cirrhosis and HCC. Clearly early liver transplantatin is potentially life saving.Your group concludes that both cholangiocarcinoma and HCC treatment in conjunction with new therapies are possible. Has your team in Romania prospectively collaborated with other center in clinical trials? Futhermore as you undoubtedly know there is growing evidence the any amount of alcohol is associated with a risk of malignancies including HCC and cholangiocarcinoma.
Reviewer 2 Report
Comments and Suggestions for Authors
Authors aimed to describe the evidence supporting expanding indications and selection criteria for liver transplantation for various oncologic
indications of primary and secondary liver tumors. This is an interesting paper.
There are several minor concerns to be addressed.
1) The reason why advanced HCC is not generally considered as liver transplantation candidate is the high probability of recurrence.
There are limited option for HCC recurrence after liver transplantation. So, the salvage treatment regimen should be summarized in the era of molecular target agent.
2) The role of so called "neoadjuvant chemo and/or radiotherapy" for downstaging should be addressed further.
In particular, the role of radiotherapy (internal or external ) combined with use of novel immune-checkpoint inhibitor should be addressed.
